# TOPIC MODELING AS MULTI-OBJECTIVE CONTRASTIVE OPTIMIZATION

**Thong Nguyen**[1], **Xiaobao Wu**[2], **Xinshuai Dong**[3], **Cong-Duy Nguyen**[2],
**See-Kiong Ng**[1], **Luu Anh Tuan**[2]*
[1]Institute of Data Science (IDS), National University of Singapore (NUS), Singapore,
[2]Nanyang Technological University (NTU), Singapore,
[3]Carnegie Mellon University (CMU), USA,

## ABSTRACT

Recent representation learning approaches enhance neural topic models by optimizing the weighted linear combination of the evidence lower bound (ELBO) of the log-likelihood and the contrastive learning objective that contrasts pairs of input documents. However, document-level contrastive learning might capture low-level mutual information, such as word ratio, which disturbs topic modeling. Moreover, there is a potential conflict between the ELBO loss that memorizes input details for better reconstruction quality, and the contrastive loss which attempts to learn topic representations that generalize among input documents. To address these issues, we first introduce a novel contrastive learning method oriented towards sets of topic vectors to capture useful semantics that are shared among a set of input documents. Secondly, we explicitly cast contrastive topic modeling as a gradient-based multi-objective optimization problem, with the goal of achieving a Pareto stationary solution that balances the trade-off between the ELBO and the contrastive objective. Extensive experiments demonstrate that our framework consistently produces higher-performing neural topic models in terms of topic coherence, topic diversity, and downstream performance.

## 1 INTRODUCTION

As one of the most prevalent methods for document analysis, topic modeling has been utilized to discover topics of document corpora, with multiple applications such as sentiment analysis (Brody & Elhadad, 2010; Tay et al., 2017; Wu & Li, 2019; Zhao et al., 2020; Nguyen et al., 2023), language generation (Jelodar et al., 2019; Nguyen et al., 2021; Tuan et al., 2020), and recommendation system (Cao et al., 2017; Zhu et al., 2017; Tay et al., 2018; Jelodar et al., 2019). In recent years, Variational Autoencoder (VAE) (Kingma & Welling, 2013) has achieved great success in many fields, and its encoder-decoder architecture has been inherited for topic modeling with neural networks, dubbed as Neural Topic Model (NTM) (Miao et al., 2016; Wu et al., 2020b; 2021; 2024a; 2023c). Exploiting the standard Gaussian as the prior distribution, neural topic models have not only produced expressive global semantics but also achieved high degree of flexibility and scalability to large-scale document collections (Wang et al., 2021b; 2020; Gupta et al., 2020).

Based on the VAE architecture, a neural topic model is trained by optimizing the evidence lower bound (ELBO), which consists of an input reconstruction loss and the KL-divergence between the prior and the topic distribution. Whereas the first component aims to optimize the reconstruction quality, the second one pressures the reconstructor by regularizing the topic representations with distributional constraints. Thus, the joint optimization induces a trade-off between the two components (Lin et al., 2019a; Asperti & Trentin, 2020). One solution to reach a balance of the trade-off is to search for the optimal weight to combine the two losses, which requires exhaustive and computationally expensive manual hyperparameter tuning (Rybkin et al., 2021). To circumvent the problem, previous works jointly train the NTM with the contrastive learning objective (Le & Akoglu, 2019; Nguyen & Luu, 2021; Li et al., 2022), since contrastive estimation has proven to be an effective regularizer to promote generalizability in input representations (Tang et al., 2021; Kim et al., 2021).

---

*Corresponding Author

Figure 1: Illustration of the intense low-level feature influence on produced topics. We record the cosine similarity of the *input* with the *document instance 1* and *document instance 2*'s topic representations generated by NTM+CL (Nguyen & Luu, 2021) and our neural topic model. Although the *input* and *instance 2* both describe the *electric* topic, the similarity for the (*input*, *instance 1*) pair is higher, because the *input* and *instance 1* have the same number of non-zero entries, *i.e.* 5, and the same ratio of maximum to minimum frequency. *i.e.* 10 to 4.

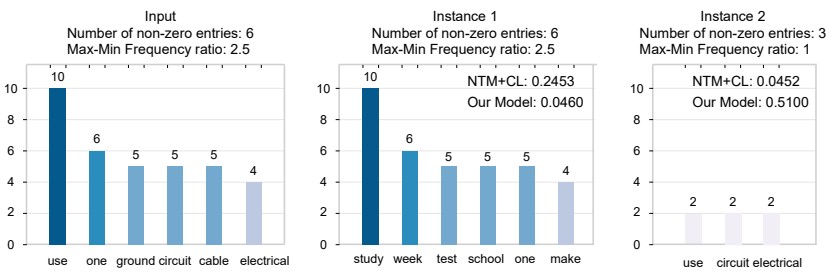

Table 1: Illustration of the effects of peculiar words on topic representations. We record the cosine similarity of the *input* and the *document instances*' topic representations generated by NTM+CL (Nguyen & Luu, 2021) and our neural topic model. As shown with underlined numbers, inserting unusual term, such as *zeppelin* or *scardino*, unexpectedly raises the similarity of the input with the document instance, even surpassing the similarity of the semantically close pair.

| Input | Document Instance | Cosine Similarity | |
|---|---|---|---|
| | | NTM+CL | Our Model |
| shuttle lands on planet | job career ask development | 0.0093 | 0.0026 |
| | star astronaut planet light moon | 0.8895 | 0.9178 |
| shuttle lands on planet *zeppelin/scardino* | job career ask development *zeppelin/scardino* | 0.9741/0.9413 | 0.0064/0.0080 |
| | star astronaut planet light moon | 0.1584/0.2547 | 0.8268/0.7188 |

However, by discriminating different document instances, contrastive learning leads to two significant challenges for neural topic models. First of all, instance discrimination could intensely concentrate on low-level mutual features (Tschannen et al., 2019), *e.g.* the number of words with non-zero frequency and the ratio of maximum-minimum word frequency, which are illustrated in Figure 1 to negatively affect topic representations. Second, as the contrastive objective bears a regularizing impact to improve generalization of hidden representations, generalizable but inefficient features to topic modeling could emerge, especially when the amount of mutual information exceeds the minimal sufficient statistics (Tian et al., 2020). For example, the discrimination task could go beyond mutual topic themes to encode common existence of peculiar words or phrases, as depicted in Table 1. As a result, there is a need to adjust the effect of contrastive learning to better control the influence of mutual information on the NTM training.

To address the first issue of low-level mutual information, we propose a novel contrastive learning for sets of documents. We hypothesize that useful signals for neural topic models should be shared by multiple input documents (Hjelm et al., 2018). For example, the model should be able to detect the sport theme in several documents in order to encode the theme into the topic representations. Intuitively, such shared semantics specify sufficient information that forms truly useful features. To implement this idea, given a mini-batch of input documents, we divide the documents into different sets and adopt augmentation to construct two views of a set. Subsequently, we obtain topic representations of a document set by aggregating topic vectors of their documents, and pass these representations to the contrastive learning objective. To maximize the training efficiency, we further shuffle the input documents in a mini-batch before diving them into different sets in order to increase the set quantity in a training iteration.

Regarding the second issue, an apparent solution is to predetermine linear weights for the ELBO and contrastive losses. Nonetheless, in general scenarios, if the tasks conflict with each other, the straightforward linear combination is an ineffective method (Liang et al., 2021; Mahapatra & Rajan, 2020). Instead, we propose to formulate the training of contrastive NTM as a multi-objective optimization problem. The optimization takes into account the gradients of the contrastive and the ELBO loss to find the Pareto stationary solution, which optimally balances the trade-off between

the two objectives (Sener & Koltun, 2018). Our proposed approach regulates the topic encoder so that it polishes the topic representations of the NTM for better topic quality, as verified by the topic coherence and topic diversity metrics in our comprehensive experiments.

The contributions of this paper are summarized as follows:

- We propose a novel setwise contrastive learning algorithm for neural topic model to resolve the problem of learning low-level mutual information of neural topic models.
- We reformulate our setwise contrastive topic modeling as a multi-task learning problem and propose to adapt multi-objective optimization algorithm to find a Pareto solution that moderates the effects of multiple objectives on the update of neural topic model parameters.
- Extensive experiments on four popular topic modeling datasets demonstrate that our approach can enhance contrastive neural topic models in terms of topic coherence, topic diversity, and downstream performance.

## 2 RELATED WORKS

**Neural topic models (NTMs).** We consider neural topic models of (Miao et al., 2016; Srivastava & Sutton, 2017; Dieng et al., 2020; Burkhardt & Kramer, 2019; Card et al., 2018; Wu et al., 2020a; Nguyen & Luu, 2021; Wu et al., 2023a; 2024b) as the closest line of related works to ours. Recent research has further sought to enrich topic capacity of NTMs through pretrained language models (Hoyle et al., 2020; Zhang et al., 2022; Bahrainian et al., 2021; Bianchi et al., 2021) and word embeddings (Zhao et al., 2020; Xu et al., 2022; Wang et al., 2022; Wu et al., 2023b). Their approaches are to inject contextualized representations from language models pretrained on large-scale upstream datasets into NTMs being fine-tuned on topic modeling downstream ones. Different from such methods, contrastive neural topic models (Nguyen & Luu, 2021; Wu et al., 2022) mainly involve downstream datasets to polish topic representations.

**Contrastive representation learning.** Contemporary approaches follow the sample-wise approach (Nguyen et al., 2022; 2023). Several methods to elegantly generate samples comprise using the momentum encoder to estimate positive views (He et al., 2020), modifying original salient/non-salient entries to create negative/positive samples (Nguyen & Luu, 2021), etc. A surge of interest presents the cluster concept into CRL, which groups hidden features into clusters and performs discrimination among groups (Guo et al., 2022; Li et al., 2020; 2021; Caron et al., 2020; Wang et al., 2021a). Different from these works, our method operates upon the input level, directly divides raw documents into random sets, and aggregates features on-the-fly during training. (Le & Hua, 2022) also allocates raw inputs into different sets. However, their work applies one pooling function to obtain set representation. Such single pooling approach might be ineffective since we would like to maximize the similarity of even unimportant entries of a positive pair, thus using min-pooling for positive set representations, and minimize the similarity of important entries of a negative pair, thus using max-pooling. Moreover, while they target image classification tasks in computer vision, we concentrate our framework upon neural topic modeling to learn discriminative topics from document corpora.

## 3 BACKGROUND

### 3.1 NEURAL TOPIC MODELS

Neural topic model (NTM) inherits the framework of Variational Autoencoder (VAE) where latent variables are construed as topics. Supposing the linguistic corpus possesses $V$ unique words, i.e. vocabulary size, each document is represented as a word count vector $\mathbf{x} \in \mathbb{R}^V$ and a latent distribution over $T$ topics $\mathbf{z} \in \mathbb{R}^T$. The NTM makes an assumption that $\mathbf{z}$ is generated from a prior distribution $p(\mathbf{z})$ and $\mathbf{x}$ is from the conditional distribution $p_\phi(\mathbf{x}|\mathbf{z})$ parameterized by a decoder $\phi$. The goal of the model is to discover the document-topic distribution given the input. The discovery is implemented as the estimation of the posterior distribution $p(\mathbf{z}|\mathbf{x})$, which is approximated by the variational distribution $q_\theta(\mathbf{z}|\mathbf{x})$, modelled by an encoder $\theta$. Inspired by VAEs, the NTM is trained to minimize the following objective based upon the Evidence Lower BOund (ELBO):

$$\min_{\theta,\phi} \mathcal{L}_{\text{ELBO}} = -\mathbb{E}_{q_\theta(\mathbf{z}|\mathbf{x})} [\log p_\phi(\mathbf{x}|\mathbf{z})] + \mathbb{KL} [q_\theta(\mathbf{z}|\mathbf{x}) \parallel p(\mathbf{z})] \tag{1}$$

The first term denotes the expected log-likelihood or the reconstruction error, the second term the Kullback-Leibler divergence that regularizes $q_\theta(\mathbf{z}|\mathbf{x})$ to be close to the prior distribution $p(\mathbf{z})$. By

enforcing the proximity of latent distribution to the prior, the regularization term puts a constraint on the latent bottleneck to bound the reconstruction capacity, thus inducing a trade-off between the two terms in the ELBO-based objective.

## 3.2 CONTRASTIVE REPRESENTATION LEARNING

Contrastive formulation maximizes the agreement by learning similar representations for different views of the same input (called positive pairs), and ensures the disagreement via generating dissimilar representations of disparate inputs (called negative pairs). Technically, given a $B$-size batch of input documents $X = \{\mathbf{x}_1, \mathbf{x}_2, \dots, \mathbf{x}_B\}, \mathbf{x}_i \in \mathbb{R}^V$, firstly data augmentation $t \sim \mathcal{T}$ is applied to each sample $\mathbf{x}_i$ to acquire the augmented view $\mathbf{y}_i$, then form the positive sample $(\mathbf{x}_i, \mathbf{y}_i)$, and pairs that involve augmentations of distinct instances $(\mathbf{x}_i, \mathbf{y}_j), i \neq j$, become negative samples. Subsequently, a prominent approach is to train a projection function $f$ by minimizing the InfoNCE loss (Oord et al., 2018): where $f$ defines a similarity mapping $\mathbb{R}^V \times \mathbb{R}^V \to \mathbb{R}$, estimated as follows:

$$f(\mathbf{x}, \mathbf{y}) = \frac{g_\varphi(\mathbf{x})^T g_\varphi(\mathbf{y})}{\| g_\varphi(\mathbf{x}) \| \| g_\varphi(\mathbf{y}) \|} / \tau, \tag{2}$$

where $g_\varphi$ denotes the neural network parameterized by $\varphi$, $\tau$ the temperature to rescale the score from the feature similarity.

## 3.3 GRADIENT-BASED MULTI-OBJECTIVE OPTIMIZATION FOR MULTI-TASK LEARNING

A Multi-Task Learning (MTL) problem considers a tuple of $M$ tasks with a vector of non-negative losses, such that some parameters $\theta^{\text{shared}}$ are shared while others $\theta^1, \theta^2, ..., \theta^M$ are task-specific:

$$\min_{\theta^{\text{shared}}} \mathcal{L}(\theta^{\text{shared}}, \theta^1, ..., \theta^M) = (\mathcal{L}_1(\theta^{\text{shared}}, \theta^1), \mathcal{L}_2(\theta^{\text{shared}}, \theta^2), \dots, \mathcal{L}_M(\theta^{\text{shared}}, \theta^M))^T, \tag{3}$$

where $\mathcal{L}_i$ denotes the loss of the $i$-th task. In most circumstances, no single parameter set can accomplish the minimal values for all loss functions. Instead, the more prevailing solution is to seek the Pareto stationary solution, which sustains optimal trade-offs among objectives (Sener & Koltun, 2018; Mahapatra & Rajan, 2020). Theoretically, a Pareto stationary solution satisfies the Karush-Kuhn-Tucker (KKT) conditions as follows,

**Theorem 1.** *(Sener & Koltun, 2018) Let $\theta^*_{shared}$ denote a Pareto point. Then, there exists non-negative scalars $\{\alpha_m\}_{m=1}^M, m \in \{1, 2, \dots, M\}$ such that :*

$$\sum_{m=1}^M \alpha_m = 1 \quad and \quad \sum_{m=1}^M \alpha_m \nabla_{\theta_{shared}} \mathcal{L}_m(\theta^*_{shared}, \theta^i) = 0. \tag{4}$$

In consequence, the conditions lead to the following optimization problem:

$$\min_{\{\alpha_m\}_{m=1}^M} \left\{ \left\| \sum_{m=1}^M \alpha_m \nabla_{\theta^{\text{shared}}} \mathcal{L}_m(\theta^{\text{shared}}, \theta^m) \right\|_2^2 \, \middle| \, \sum_{m=1}^M \alpha_m = 1, \alpha_m \geq 0 \quad \forall m \right\} \tag{5}$$

Either the solution to the above problem does not exist, *i.e.* the outcome meets the KKT conditions, or the solution supplies a descent direction that decreases all per-task losses (Désidéri, 2012). Hence, the procedure would become gradient descent on task-oriented parameters followed by solving Eq. (5) and adapting $\sum_{m=1}^M \alpha_m \nabla_{\theta^{\text{shared}}} \mathcal{L}_m(\theta^{\text{shared}}, \theta^m)$ as the gradient update upon shared parameters.

## 4 METHODOLOGY

In this section, we introduce the details and articulate the analysis of our proposed setwise contrastive learning approach. Thereupon, we delineate the multi-objective optimization framework for contrastive neural topic models.

## 4.1 SETWISE CONTRASTIVE LEARNING FOR NEURAL TOPIC MODEL

---

**Algorithm 1** Setwise Contrastive Neural Topic Model as Multi-Objective Optimization.

---

**Input:** Document batch $\mathcal{X} = \{\mathbf{x}_1, \mathbf{x}_2, \ldots, \mathbf{x}_B\}$, encoder $h_\theta$, decoder $g_\phi$; Positive augmentation $t^+$, negative augmentation $t^-$; Learning rate $\eta$; Set size $K$; contrastive weight $\beta$; Shuffling number $S$;

1: **for** $i = 1$ to $B$ **do**
2: $\quad$ $\mathbf{x}_i^+ = t^+(\mathbf{x}_i), \mathbf{x}_i^- = t^-(\mathbf{x}_i)$;
3: $\quad$ $\mathbf{z}_i = h_\theta(\mathbf{x}_i), \mathbf{z}_i^+ = h_\theta(\mathbf{x}_i^+), \mathbf{z}_i^- = h_\theta(\mathbf{x}_i^-)$;
4: **end for**
5: *# Matrix $M$ stores $S$ shuffled rows of indices $1, 2, \ldots, B$*
6: $M$ = index matrix of size $S \times B$;
7: **for** $s = 1$ to $S$ **do**
8: $\quad$ *# Group every $K$ documents $jK + 0, jK + 1, \ldots, jK + (K-1)$ into a set. Different shuffled row $s$ leads to different document sets.*
9: $\quad$ **for** $j = 1$ to $\frac{B}{K}$ **do**
10: $\quad\quad$ *# Feature extraction for document sets. Refer to Section 5.3 and Appendix C for choices of pooling functions $\varphi^-$ and $\varphi^+$, respectively.*
11: $\quad\quad$ $\mathbf{s}_{s,j}^{\varphi^-} = \varphi^-\left(\{\mathbf{z}_{s,jK+i}\}_{k=0}^{K-1}\right)$;
12: $\quad\quad$ $\mathbf{s}_{s,j}^{\varphi^+} = \varphi^+\left(\{\mathbf{z}_{s,jK+i}\}_{k=0}^{K-1}\right)$;
13: $\quad\quad$ $\mathbf{s}_{s,j}^- = \varphi^-\left(\{\mathbf{z}_{s,jK+i}^-\}_{k=0}^{K-1}\right)$;
14: $\quad\quad$ $\mathbf{s}_{s,j}^+ = \varphi^+\left(\{\mathbf{z}_{s,jK+i}^+\}_{k=0}^{K-1}\right)$;
15: $\quad$ **end for**
16: **end for**
17: *# Setwise contrastive learning objective*
18: $\mathcal{L}_{\text{InfoNCE}} = -\sum\limits_{s=1}^{S} \sum\limits_{j=1}^{\frac{B}{K}} \log \dfrac{\exp(f(\mathbf{s}_{s,j}^{\varphi^+}, \mathbf{s}_{s,j}^+))}{\exp(f(\mathbf{s}_{s,j}^{\varphi^+}, \mathbf{s}_{s,j}^+)) + \sum\limits_{q=1}^{S} \sum\limits_{t=1}^{\frac{B}{K}} \mathbb{1}[(s,j) \neq (q,t)] \exp(f(\mathbf{s}_{s,j}^{\varphi^-}, \mathbf{s}_{q,t}^-))}$
19: *# Multi-objective optimization. Refer to Eq. 8 for $\alpha$ formulation*
20: $\alpha = \text{solver}[\nabla_\theta \mathcal{L}_{\text{InfoNCE}}(\theta), \nabla_\theta \mathcal{L}_{\text{ELBO}}(\theta, \phi)]$;
21: $G_\theta = \alpha \nabla_\theta \mathcal{L}_{\text{InfoNCE}} + (1 - \alpha) \nabla_\theta \mathcal{L}_{\text{ELBO}}$;
22: $G_\phi = \nabla \mathcal{L}_{\text{ELBO}}$;
23: *# Update topic encoder $\theta$ and topic decoder $\phi$*
24: $\theta = \theta - \eta \cdot G_\theta$;
25: $\phi = \phi - \eta \cdot G_\phi$;

---

**Sample augmentation.** Firstly, given a batch of $B$ input documents $X = \{\mathbf{x}_1, \mathbf{x}_2, \ldots, \mathbf{x}_B\}$, we conduct data augmentation to generate the corresponding $2B$ documents $\mathcal{X}^+ = \{\mathbf{x}_1^+, \mathbf{x}_2^+, \ldots, \mathbf{x}_B^+\}$ and $\mathcal{X}^- = \{\mathbf{x}_1^-, \mathbf{x}_2^-, \ldots, \mathbf{x}_B^-\}$, where $\mathbf{x}_i^+$ and $\mathbf{x}_i^-$ are semantically close to and distant from $\mathbf{x}_i$, respectively. Previous methods (Yin & Schütze, 2016; Nguyen & Luu, 2021) utilize pre-trained word embeddings or auxiliary metric such as TF-IDF to generate such positive and negative samples. However, those approaches only modify certain words, which are shown to produce noisy and not true positive/negative documents (Vizcarra & Ochoa-Luna, 2020). Therefore, with the advance of LLMs, before forming the BoW vector of each input document, we adopt the ChatGPT API to perform the augmentation with the prompt template: *A sentence that is <related/unrelated> to this text: <input document>*, and use the *<related>* option for $\mathcal{X}^+$ and the *<unrelated>* one for $\mathcal{X}^-$. As shown in Appendix E, our approach produces document augmentations that are more natural and less noisy compared with previous methods.

**Feature extraction.** For a mini-batch, each set is constructed by grouping every $K$ input samples. We proceed to extract set representation:

$$\mathbf{s}^{\varphi^-} = \varphi^-(\{h_\theta(\mathbf{x}_i)\}_{k=0}^{K-1}), \quad \mathbf{s}^{\varphi^+} = \varphi^+(\{h_\theta(\mathbf{x}_i)\}_{k=0}^{K-1}), \quad \mathbf{s}^- = \varphi^-(\{h_\theta(\mathbf{x}_i^-)\}_{k=0}^{K-1}), \quad \mathbf{s}^+ = \varphi^+(\{h_\theta(\mathbf{x}_i^+)\}_{k=0}^{K-1}), \tag{6}$$

where $h_\theta$ denotes the $\theta$-parameterized topic encoder. $\varphi^-$ and $\varphi^+$ are pooling functions for positive and negative set, respectively. We experiment with different choices for $\varphi^-$ and $\varphi^+$ in Section 5.3 and Appendix C, respectively.

**Index shuffling.** To create sets of $K$ documents, we can allocate every $K$ document into a set. However, this approach results in a restricted quantity of document sets, *i.e.* $\lfloor \frac{B}{K} \rfloor$ sets. Consequently, this yields $\lfloor \frac{B}{K} \rfloor$ positive and $2\left(\lfloor \frac{B}{K} \rfloor - 1\right)$ negative pairs for each mini-batch. This method is also not efficient in terms of document input usage, as each document only appears once in a certain set.

Instead, to augment the number of documents and consequently raise the number of positive and negative set pairs, we shuffle the list of indices from 1 to $B$ for $S$ times and record the shuffled indices in an index matrix $S$ of size $S \times B$. We then assemble every $K$ elements corresponding to

the index matrix into a document set. Our proposed approach can increase the number of positive and negative pairs through raising the set number to $\lfloor \frac{BS}{K} \rfloor$ sets, thus maximizing the capacity of our setwise contrastive neural topic model.

## 4.2 Multi-Objective Optimization for Neural Topic Model

**Problem statement.** For a given neural topic model, we seek to find preference vectors $\boldsymbol{\alpha}$ that produces adept gradient update policy $\sum_i \alpha_i \nabla f_i$. For the contrastive neural topic model, because only the neural topic encoder receives the gradients of both contrastive and topic modeling objectives, we focus on modulating the encoder update while keeping the decoder learning intact. Hence, motivated by formulation (5), our optimization problem becomes:

$$\min_{\alpha} \left\{ \left\| \alpha \nabla_\theta \mathcal{L}_{\text{InfoNCE}}(\theta) + (1-\alpha)\nabla_\theta \mathcal{L}_{\text{ELBO}}(\theta, \phi) \right\|_2^2 \Big| \alpha \geq 0 \right\} \tag{7}$$

which is a convex quadratic optimization with linear constraints. Erstwhile research additionally integrates prior information to compose solutions that are preference-specific among tasks (Mahapatra & Rajan, 2020; Lin et al., 2019b; 2020). However, for neural topic modeling domain, such information is often unavailable, so we exclude it from our problem notation.

**Optimization solution.** Our problem formulation involves convex optimization of multi-dimensional quadratic function with linear constraints. Formally, we derive the analytical solution $\hat{\alpha}$ as:

$$\hat{\alpha} = \text{solver}[\nabla_\theta \mathcal{L}_{\text{InfoNCE}}(\theta), \nabla_\theta \mathcal{L}_{\text{ELBO}}(\theta, \phi)] = - \left[ \frac{(\nabla_\theta \mathcal{L}_{\text{InfoNCE}}(\theta) - \nabla_\theta \mathcal{L}_{\text{ELBO}}(\theta, \phi))^\top \nabla_\theta \mathcal{L}_{\text{InfoNCE}}(\theta)}{\left\| \nabla_\theta \mathcal{L}_{\text{InfoNCE}}(\theta) - \nabla_\theta \mathcal{L}_{\text{ELBO}}(\theta, \phi) \right\|_2^2} \right]_+ \tag{8}$$

where $[.]_+$ denotes the ReLU operation. After accomplishing $\hat{\alpha}$, we plug the vector to control the gradient for the neural topic encoder, while maintaining the update upon the neural topic decoder whose value is determined by the back-propagation of the reconstruction loss. Our multi-objective framework for setwise contrastive neural topic model is depicted in Algorithm 1.

## 5 Experiments

In this section, we conduct experiments and empirically demonstrate the effectiveness of the proposed methods for NTMs. We provide the experimental setup and report numerical results along with qualitative studies.

### 5.1 Experimental setup

**Benchmark datasets.** We adopt popular benchmark datasets spanning various domains, vocabulary sizes, and document lengths for experiments: (i) **20Newsgroups (20NG)** (Lang, 1995), one of the most well-known datasets for topic model evaluation, consisting of 18000 documents with 20 labels; (ii) **IMDb** (Maas et al., 2011), the dataset of movie reviews, belonging to two sentiment labels, i.e. positive and negative; (iii) **Wikitext-103 (Wiki)** (Merity et al., 2016), comprising 28500 articles from the Good and Featured section on Wikipedia; (iv) **AG News** (Zhang et al., 2015), consisting of news titles and articles whose size is 30000 and 1900 for training and testing subsets, respectively.

**Evaluation metrics.** Following previous mainstream works (Hoyle et al., 2020; Wang et al., 2019; Card et al., 2018; Nguyen & Luu, 2021; Xu et al., 2022; Wu et al., 2023b; Zhao et al., 2020; Wang et al., 2022), we evaluate our topic models concerning topic coherence and topic diversity. In particular, we extract the top 10 words in each topic, then utilize the testing split of each dataset as the reference corpus to measure the topics' Normalized Pointwise Mutual Information (NPMI) (Röder et al., 2015) in terms of topic coherence, and the Topic Diversity (TD) metric (Röder et al., 2015) in terms of topic diversity. Furthermore, because topic quality also depends upon the usefulness for downstream applications, we evaluate the document classification performance of the produced topics and use F1 score as the quality measure.

Table 2: Topic Coherence and Topic Diversity results on 20NG and IMDb datasets.

| Method | 20NG | | | | IMDb | | | |
| --- | --- | --- | --- | --- | --- | --- | --- | --- |
| | $T = 50$ | | $T = 200$ | | $T = 50$ | | $T = 200$ | |
| | NPMI | TD | NPMI | TD | NPMI | TD | NPMI | TD |
| NTM | $0.283_{\pm0.004}$ | $0.734_{\pm0.009}$ | $0.277_{\pm0.003}$ | $0.686_{\pm0.004}$ | $0.170_{\pm0.008}$ | $0.777_{\pm0.021}$ | $0.169_{\pm0.003}$ | $0.690_{\pm0.015}$ |
| ETM | $0.305_{\pm0.006}$ | $0.776_{\pm0.022}$ | $0.264_{\pm0.002}$ | $0.623_{\pm0.002}$ | $0.174_{\pm0.001}$ | $0.805_{\pm0.019}$ | $0.168_{\pm0.001}$ | $0.687_{\pm0.007}$ |
| DVAE | $0.320_{\pm0.005}$ | $0.824_{\pm0.017}$ | $0.269_{\pm0.003}$ | $0.786_{\pm0.005}$ | $0.183_{\pm0.004}$ | $0.836_{\pm0.010}$ | $0.173_{\pm0.006}$ | $0.739_{\pm0.005}$ |
| BATM | $0.314_{\pm0.003}$ | $0.786_{\pm0.014}$ | $0.245_{\pm0.001}$ | $0.623_{\pm0.008}$ | $0.065_{\pm0.008}$ | $0.619_{\pm0.016}$ | $0.090_{\pm0.004}$ | $0.652_{\pm0.008}$ |
| W-LDA | $0.279_{\pm0.003}$ | $0.719_{\pm0.026}$ | $0.188_{\pm0.001}$ | $0.614_{\pm0.002}$ | $0.136_{\pm0.007}$ | $0.692_{\pm0.016}$ | $0.095_{\pm0.003}$ | $0.666_{\pm0.009}$ |
| SCHOLAR | $0.319_{\pm0.007}$ | $0.788_{\pm0.008}$ | $0.263_{\pm0.002}$ | $0.634_{\pm0.006}$ | $0.168_{\pm0.002}$ | $0.702_{\pm0.014}$ | $0.140_{\pm0.001}$ | $0.675_{\pm0.005}$ |
| SCHOLAR + BAT | $0.324_{\pm0.006}$ | $0.824_{\pm0.011}$ | $0.272_{\pm0.002}$ | $0.648_{\pm0.009}$ | $0.182_{\pm0.002}$ | $0.825_{\pm0.008}$ | $0.175_{\pm0.003}$ | $0.761_{\pm0.010}$ |
| NTM+CL | $0.332_{\pm0.006}$ | $0.853_{\pm0.005}$ | $0.277_{\pm0.003}$ | $0.699_{\pm0.004}$ | $0.191_{\pm0.004}$ | $0.857_{\pm0.010}$ | $0.186_{\pm0.002}$ | $0.843_{\pm0.008}$ |
| HyperMiner | $0.305_{\pm0.005}$ | $0.613_{\pm0.023}$ | $0.254_{\pm0.002}$ | $0.646_{\pm0.004}$ | $0.182_{\pm0.004}$ | $0.485_{\pm0.009}$ | $0.177_{\pm0.002}$ | $0.658_{\pm0.012}$ |
| WeTe | $0.304_{\pm0.005}$ | $0.749_{\pm0.018}$ | $0.254_{\pm0.001}$ | $0.742_{\pm0.005}$ | $0.167_{\pm0.004}$ | $0.831_{\pm0.010}$ | $0.163_{\pm0.005}$ | $0.738_{\pm0.008}$ |
| TSCTM | $0.271_{\pm0.007}$ | $0.668_{\pm0.019}$ | $0.226_{\pm0.001}$ | $0.662_{\pm0.006}$ | $0.149_{\pm0.003}$ | $0.741_{\pm0.008}$ | $0.145_{\pm0.002}$ | $0.658_{\pm0.012}$ |
| **Our model** | $\mathbf{0.340}_{\pm0.005}$ | $\mathbf{0.913}_{\pm0.019}$ | $\mathbf{0.291}_{\pm0.003}$ | $\mathbf{0.905}_{\pm0.004}$ | $\mathbf{0.200}_{\pm0.007}$ | $\mathbf{0.916}_{\pm0.008}$ | $\mathbf{0.197}_{\pm0.003}$ | $\mathbf{0.892}_{\pm0.007}$ |

Table 3: Topic Coherence and Topic Diversity results on Wiki and AG News datasets.

| Method | Wiki | | | | AG News | | | |
| --- | --- | --- | --- | --- | --- | --- | --- | --- |
| | $T = 50$ | | $T = 200$ | | $T = 50$ | | $T = 200$ | |
| | NPMI | TD | NPMI | TD | NPMI | TD | NPMI | TD |
| NTM | $0.250_{\pm0.010}$ | $0.817_{\pm0.006}$ | $0.291_{\pm0.009}$ | $0.624_{\pm0.011}$ | $0.197_{\pm0.015}$ | $0.729_{\pm0.007}$ | $0.205_{\pm0.002}$ | $0.797_{\pm0.002}$ |
| ETM | $0.332_{\pm0.003}$ | $0.756_{\pm0.013}$ | $0.317_{\pm0.009}$ | $0.671_{\pm0.006}$ | $0.204_{\pm0.004}$ | $0.736_{\pm0.006}$ | $0.055_{\pm0.002}$ | $0.683_{\pm0.005}$ |
| DVAE | $0.404_{\pm0.006}$ | $0.815_{\pm0.009}$ | $0.359_{\pm0.007}$ | $0.721_{\pm0.011}$ | $0.271_{\pm0.016}$ | $0.850_{\pm0.013}$ | $0.182_{\pm0.001}$ | $0.746_{\pm0.005}$ |
| BATM | $0.336_{\pm0.010}$ | $0.807_{\pm0.005}$ | $0.319_{\pm0.005}$ | $0.732_{\pm0.012}$ | $0.256_{\pm0.019}$ | $0.754_{\pm0.008}$ | $0.144_{\pm0.003}$ | $0.711_{\pm0.006}$ |
| W-LDA | $0.451_{\pm0.007}$ | $0.836_{\pm0.007}$ | $0.308_{\pm0.011}$ | $0.725_{\pm0.014}$ | $0.270_{\pm0.033}$ | $0.844_{\pm0.014}$ | $0.135_{\pm0.004}$ | $0.708_{\pm0.004}$ |
| SCHOLAR | $0.429_{\pm0.011}$ | $0.821_{\pm0.010}$ | $0.446_{\pm0.009}$ | $0.860_{\pm0.014}$ | $0.278_{\pm0.024}$ | $0.886_{\pm0.007}$ | $0.185_{\pm0.004}$ | $0.747_{\pm0.002}$ |
| SCHOLAR + BAT | $0.446_{\pm0.010}$ | $0.824_{\pm0.006}$ | $0.455_{\pm0.007}$ | $0.854_{\pm0.011}$ | $0.272_{\pm0.027}$ | $0.859_{\pm0.008}$ | $0.189_{\pm0.003}$ | $0.750_{\pm0.001}$ |
| NTM+CL | $0.481_{\pm0.005}$ | $0.841_{\pm0.012}$ | $0.462_{\pm0.016}$ | $0.831_{\pm0.016}$ | $0.279_{\pm0.025}$ | $0.890_{\pm0.007}$ | $0.190_{\pm0.002}$ | $0.790_{\pm0.006}$ |
| HyperMiner | $0.344_{\pm0.005}$ | $0.807_{\pm0.008}$ | $0.446_{\pm0.007}$ | $0.844_{\pm0.012}$ | $0.259_{\pm0.017}$ | $0.821_{\pm0.013}$ | $0.180_{\pm0.001}$ | $0.743_{\pm0.001}$ |
| WeTe | $0.367_{\pm0.004}$ | $0.845_{\pm0.012}$ | $0.453_{\pm0.007}$ | $0.827_{\pm0.005}$ | $0.283_{\pm0.006}$ | $0.878_{\pm0.012}$ | $0.186_{\pm0.003}$ | $0.754_{\pm0.006}$ |
| TSCTM | $0.327_{\pm0.006}$ | $0.753_{\pm0.010}$ | $0.404_{\pm0.008}$ | $0.737_{\pm0.012}$ | $0.252_{\pm0.026}$ | $0.783_{\pm0.008}$ | $0.166_{\pm0.002}$ | $0.672_{\pm0.004}$ |
| **Our model** | $\mathbf{0.496}_{\pm0.010}$ | $\mathbf{0.959}_{\pm0.010}$ | $\mathbf{0.491}_{\pm0.006}$ | $\mathbf{0.938}_{\pm0.011}$ | $\mathbf{0.351}_{\pm0.012}$ | $\mathbf{0.934}_{\pm0.012}$ | $\mathbf{0.221}_{\pm0.003}$ | $\mathbf{0.885}_{\pm0.004}$ |

**Baseline models.** We compare our proposed approaches with the following state-of-the-art NTMs: (i) **(NTM)** (Miao et al., 2016), inheriting the encoder-decoder paradigm of the VAE architecture and standard Gaussian as the prior distribution; (ii) **(ETM)** (Dieng et al., 2020), modeling the topic-word distribution with topic and word embeddings; (iii) **(DVAE)** (Burkhardt & Kramer, 2019), whose Dirichlet is the prior for both topic and word distributions; (iv) **BATM** (Wang et al., 2020), which is based on GAN to consist of an encoder, a generator, and a discriminator; (v) **W-LDA** (Nan et al., 2019), which utilizes Wasserstein autoencoder as a topic model; (vi) **SCHOLAR** (Card et al., 2018), which incorporates external variables and logistic normal as the prior distribution; (vii) **SCHOLAR + BAT** (Hoyle et al., 2020), a SCHOLAR model applying knowledge distillation in which BERT is the teacher and NTM is the student; (viii) **NTM + CL** (Nguyen & Luu, 2021), which jointly trains evidence lower bound and the individual-based contrastive learning objective; (ix) **HyperMiner** (Xu et al., 2022), which adopts embeddings in the hyperbolic space to model topics; (x) **WeTe** (Wang et al., 2022), which replaces the reconstruction error with the conditional transport distance; (xi) **TSCTM** (Wu et al., 2022): A contrastive NTM which uses an auxiliary semantic space to generate positive and negative samples for short text topic modeling.

**Implementation details.** We evaluate our approach both at $T = 50$ and $T = 200$ topics. We experiment with different set cardinality $K \in \{1, 2, 3, 4, 5, 6\}$ and permutation matrix size $P \in \{1, 2, 4, 8, 16, 32\}$. More implementation details can be found in Appendix A.

## 5.2 MAIN RESULTS

**Overall results.** Table 2 and 3 show the topic quality results of our proposed method and the baselines. We run all models five times with different random seeds and record the mean and standard deviation of the results. Our method effectively produces topics that are of high quality in terms of topic diversity and topic coherence. In all four datasets, lower TD scores of baseline methods imply that they generate repetitive and less diverse topics than ours. For example, on 20NG for $T = 50$ our TD scores all surpass NTM+CL, HyperMiner, and WeTe (0.959 v.s. 0.841, 0.807, and 0.845). With regards to topic coherence, we completely outperform all these methods on NPMI (*e.g.* on Wiki and $T = 200$, 0.491 v.s. 0.462, 0.446, and 0.453). We also conduct significance tests and observe that the p-values are all smaller than 0.05, which verifies the significance of our improvements. As

Table 4: Document classification results with topic representations.

| Method | 20NG | IMDb | AG News |
|---|---|---|---|
| BATM | 30.8 | 66.0 | 56.0 |
| SCHOLAR | 32.2 | 83.4 | 74.8 |
| SCHOLAR + BAT | 52.9 | 73.1 | 59.0 |
| NTM+CL | 54.4 | 84.2 | 76.2 |
| HyperMiner | 43.0 | 71.3 | 63.4 |
| WeTe | 48.5 | 80.3 | 65.7 |
| **Our Model** | **57.0** | **86.4** | **78.6** |

Table 5: Ablation Results on the Objective Control Strategy.

| Method | Wiki | |
|---|---|---|
| | $T = 50$ | $T = 200$ |
| UW | 0.482±0.009 | 0.474±0.005 |
| GradNorm | 0.485±0.007 | 0.469±0.006 |
| PCGrad | 0.489±0.011 | 0.472±0.005 |
| RW | 0.492±0.009 | 0.476±0.007 |
| Linear-$\alpha = 0.25$ | 0.483±0.011 | 0.487±0.007 |
| Linear-$\alpha = 0.50$ | 0.491±0.015 | 0.485±0.004 |
| **Our Model** | **0.496±0.010** | **0.491±0.006** |

Figure 2: (left) Jensen-Shannon for aligned topic pairs of NTM+CL (Nguyen & Luu, 2021) and Our Model. (right) The number of aligned topic pairs which Our Model improves upon NTM+CL (Nguyen & Luu, 2021).

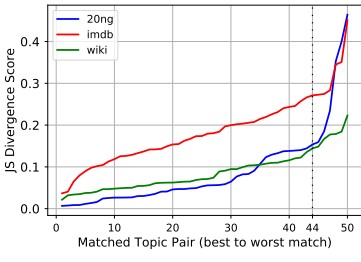 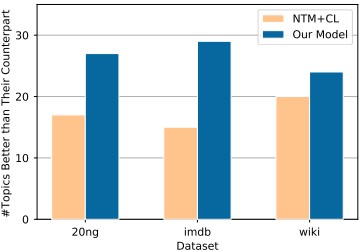

such, these results demonstrate the influence of the useful mutual information brought by our setwise contrastive learning method and the balance between the contrastive and ELBO objectives.

**Topic-by-topic results.** We perform individual comparison upon each of our topics with the aligned one generated by the baseline contrastive NTM. Inspired by (Hoyle et al., 2020), we construct a bipartite graph connecting our topics with the baseline ones. We adopt competitive linking to greedily estimate the optimal weight for the matching in our bipartite graph matching. The weight of every connection is the Jensen-Shannon (JS) divergence between two topics. Each iteration will retrieve two topics whose JS score is the lowest and proceed to remove them from the topic list. The procedure is repeated until the lowest JS score surpasses a definite threshold. Figure 2 (left) depicts the aligned score for three benchmark datasets. Inspecting visually, we extract 44 most aligned topic pairs for the comparison procedure. As shown in Figure 2 (right), we observe that our approach possesses more high-NPMI topics then the NTM+CL baseline. This demonstrates the setwise contrastive method not only improves general topic quality but also manufactures better individual topics.

**Document classification.** We evaluate the effectiveness of our topic representations for the downstream task. Particularly, we extract latent distributions inferred by NTMs in the $T = 50$ setting and train Random Forest classifiers with the number of decision trees of 100 to predict the category of each document. Because the Wiki dataset does not include a corresponding label to each input document, we use 20NG, IMDb, and AGNews for this experiment. As shown in Table 4, our model accomplishes the best performance over other NTMs, substantiating the refined usefulness of the topic representations yielded by our model.

### 5.3 ABLATION STUDY

**Objective control strategy.** We perform an ablation study and analyze the importance of our proposed multi-objective training framework. In detail, we remove the Pareto stationary solver and then conduct the training with the approaches of linear combination, in which we attach the weight $\alpha$ to the contrastive objective with $\alpha \in \{0.25, 0.5\}$, the Uncertainty Weighting (UW) (Kendall et al., 2018), gradient normalization (GradNorm) (Chen et al., 2018), Projecting Conflicting Gradients (PCGrad) (Yu et al., 2020), and Random Weighting (RW) (Lin et al., 2021). Without the multi-objective framework, the contrastive NTM suffers from sub-optimal performance, as shown in Table 5. Hypothetically, this means that in the fixed linear and heuristics weighting systems, the optimization procedure could not efficiently adapt to the phenomenon that the contrastive objective overwhelms the topic modeling task. Hence, contrasting samples might supply excessive mutual information

Table 6: Topic Coherence results with different pooling functions for negative set representation.

| Method | 20NG | | IMDb | | Wiki | | AGNews | |
|---|---|---|---|---|---|---|---|---|
| | $T = 50$ | $T = 200$ | $T = 50$ | $T = 200$ | $T = 50$ | $T = 200$ | $T = 50$ | $T = 200$ |
| MinPooling | $0.332_{\pm 0.003}$ | $0.280_{\pm 0.005}$ | $0.190_{\pm 0.005}$ | $0.186_{\pm 0.004}$ | $0.491_{\pm 0.007}$ | $0.481_{\pm 0.004}$ | $0.318_{\pm 0.015}$ | $0.210_{\pm 0.002}$ |
| MeanPooling | $0.334_{\pm 0.005}$ | $0.283_{\pm 0.005}$ | $0.194_{\pm 0.006}$ | $0.195_{\pm 0.010}$ | $0.492_{\pm 0.006}$ | $0.484_{\pm 0.009}$ | $0.321_{\pm 0.014}$ | $0.211_{\pm 0.002}$ |
| SumPooling | $0.333_{\pm 0.004}$ | $0.281_{\pm 0.003}$ | $0.191_{\pm 0.007}$ | $0.188_{\pm 0.005}$ | $0.493_{\pm 0.013}$ | $0.484_{\pm 0.004}$ | $0.322_{\pm 0.006}$ | $0.215_{\pm 0.003}$ |
| GAP | $0.339_{\pm 0.004}$ | $0.288_{\pm 0.004}$ | $0.197_{\pm 0.003}$ | $0.196_{\pm 0.005}$ | $0.495_{\pm 0.009}$ | $0.485_{\pm 0.007}$ | $0.323_{\pm 0.007}$ | $0.221_{\pm 0.003}$ |
| FSP | $0.339_{\pm 0.007}$ | $0.286_{\pm 0.003}$ | $0.199_{\pm 0.004}$ | $0.195_{\pm 0.007}$ | $0.494_{\pm 0.012}$ | $0.484_{\pm 0.007}$ | $0.324_{\pm 0.012}$ | $0.221_{\pm 0.003}$ |
| PSWE | $0.334_{\pm 0.006}$ | $0.284_{\pm 0.004}$ | $0.194_{\pm 0.006}$ | $0.191_{\pm 0.003}$ | $0.494_{\pm 0.013}$ | $0.484_{\pm 0.009}$ | $0.323_{\pm 0.006}$ | $0.218_{\pm 0.001}$ |
| **MaxPooling** | $\mathbf{0.340}_{\pm 0.005}$ | $\mathbf{0.291}_{\pm 0.003}$ | $\mathbf{0.200}_{\pm 0.007}$ | $\mathbf{0.197}_{\pm 0.003}$ | $\mathbf{0.496}_{\pm 0.010}$ | $\mathbf{0.491}_{\pm 0.006}$ | $\mathbf{0.351}_{\pm 0.012}$ | $\mathbf{0.221}_{\pm 0.003}$ |

Table 7: Examples of the topics produced by NTM+CL (Nguyen & Luu, 2021) and Our Model.

| Dataset | Method | NPMI | Topic |
|---|---|---|---|
| 20NG | NTM+CL | 0.2766 | mouse monitor orange gateway video apple screen card port vga |
| | Our Model | 0.3537 | vga monitor monitors colors video screen card mhz cards color |
| IMDb | NTM+CL | 0.1901 | seagal ninja martial arts zombie zombies jet fighter flight helicopter |
| | Our Model | 0.3143 | martial arts seagal jackie chan kung hong ninja stunts kong |
| Wiki | NTM+CL | 0.1070 | architectural castle architect buildings grade historic coaster roller sculpture tower |
| | Our Model | 0.2513 | century building built church house site castle buildings historic listed |

which eclipses the useful features for topic learning. Due to the length limit, more experiments upon the objective control strategy can be referred to Appendix B.

**Pooling functions.** We experiment with different pooling operations for negative samples $\varphi^-$ and compare their performance with our choice of MaxPooling. The experiment of pooling functions for positive samples $\varphi^+$ can be found in Appendix C. We adapt MeanPooling, SumPooling, Global Attention Pooling (GAP) (Li et al., 2016), Featurewise Sort Pooling (FSP) (Zhang et al., 2018), and Pooling by Sliced-Wasserstein Embedding (PSWE) (Naderializadeh et al., 2021). As can be observed from Table 6, MaxPooling acquires the highest topic coherence quality. We hypothesize that different from other aggregation functions, MaxPooling directly retrieves strong features, thus being able to extract important information and suppress the noise. We also realize that our proposed framework consistently outperforms the individual-based contrastive NTM, demonstrating the robustness of setwise contrastive learning with a variety of aggregation functions.

### 5.4 ANALYSIS

**Effects of setwise and instance-based contrastive learning on topic representations.** Table 1 and Figure 1 include similarity scores of the input with different document samples. Different from NTM+CL, our model pulls together latent representations of semantically close topics. In Figure 1, *instance 2* still maintains high equivalence level, despite its noticeable distinction with the input in terms of the low-level properties. Furthermore, in Table 1, inserting unfamiliar words does not alter the similarity scale between the input and two document samples. These results indicate that our setwise contrastive learning focuses on encoding topic information and avoids impertinent low-level vector features.

**Case study.** Table 7 shows randomly selected examples of the discovered topics by the individual-based and our setwise contrastive model. In general, our model produces topics which are more coherent and less repetitive. For example, in the 20NG dataset, our inferred topic focuses on "*computer screening*" topic with closely correlated terms such as "*monitor*", "*port*", and "*vga*", whereas topic of NTM+CL consists of "*apple*" and "*orange*", which are related to "*fruits*". For IMDb, our topic mainly involves words concerning "*martial art movies*". However, NTM+CL mix those words with "*zombies*", "*flight*", and "*helicopter*". In the same vein, there exists irrelevant words within the baseline topic in the Wiki dataset, for example "*grade*", "*roller*", and "*coaster*", while our topic entirely concentrates on "*architecture*".

## 6 CONCLUSION

In this paper, we propose a novel and well-motivated Multi-objective Setwise Contrastive Neural Topic Model. Our contrastive approach incorporates the set concept to encourage the learning of common instance features and potentially more accuracy construction of hard negative samples. We propose to adapt multi-objective optimization to efficiently polish the encoded topic representations. Extensive experiments indicate that our framework accomplishes state-of-the-art performance in terms of producing both high-quality and useful topics.

# 7 ACKNOWLEDGEMENTS

This research/project is supported by the National Research Foundation, Singapore under its AI Singapore Programme (AISG Award No: AISG3-PhD-2023-08-051T). Thong Nguyen is supported by a Google Ph.D. Fellowship in Natural Language Processing.

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

# Appendix

## Table of Contents

## A HYPERPARAMETER SETTINGS

In Table 8, we denote hyperparameter details of our neural topic models, i.e. learning rate $\eta$, batch size $B$, and the temperature $\tau$ for the InfoNCE loss. For training execution, the hyperparameters vary with respect to the dataset.

Table 8: Hyperparameter Settings for Neural Topic Model Training.

| Hyperparameter | 20NG | | IMDb | | Wiki | |
|---|---|---|---|---|---|---|
| | $T = 50$ | $T = 200$ | $T = 50$ | $T = 200$ | $T = 50$ | $T = 200$ |
| sample set size $K$ | 4 | 4 | 3 | 3 | 4 | 4 |
| permutation matrix size $P$ | 8 | 8 | 8 | 8 | 8 | 8 |
| temperature $\tau$ | 0.2 | 0.2 | 0.2 | 0.2 | 0.2 | 0.2 |
| learning rate $\eta$ | 0.002 | 0.002 | 0.002 | 0.002 | 0.001 | 0.002 |
| batch size $B$ | 200 | 200 | 200 | 200 | 500 | 500 |

## B FURTHER STUDY OF OBJECTIVE CONTROL STRATEGIES

We extend the comparison of multi-objective optimization (MOO) and other control strategies on 20NG and IMDb datasets in Table 9. In general, we find that our adaptation of gradient-based MOO balances the contrastive and ELBO losses more proficiently to obtain better topic quality.

## C FURTHER STUDY OF POOLING FUNCTION FOR POSITIVE SET REPRESENTATION

In this appendix, we extend our ablation study of different choices of pooling functions $\varphi^-$ for negative set in Section 5.3 to pooling function $\varphi^+$ for positive set representation. In particular, similar to Section 5.3, we also compare our choice of pooling function for positive sets, *i.e.* MinPooling, with MaxPooling, MeanPooling, SumPooling, GAP (Li et al., 2016), FSP (Zhang et al., 2018), and PSWE (Naderializadeh et al., 2021). The results are denoted in Table 10.

Table 9: Topic Coherence and Topic Diversity results on 20NG and IMDb datasets with different objective control strategies.

| Method | 20NG | | | | IMDb | | | |
|---|---|---|---|---|---|---|---|---|
| | $T = 50$ | | $T = 200$ | | $T = 50$ | | $T = 200$ | |
| | NPMI | TD | NPMI | TD | NPMI | TD | NPMI | TD |
| UW | $0.328 \pm 0.006$ | $0.837 \pm 0.011$ | $0.273 \pm 0.003$ | $0.831 \pm 0.004$ | $0.187 \pm 0.004$ | $0.848 \pm 0.007$ | $0.175 \pm 0.009$ | $0.818 \pm 0.007$ |
| GradNorm | $0.338 \pm 0.006$ | $0.870 \pm 0.003$ | $0.276 \pm 0.005$ | $0.839 \pm 0.006$ | $0.189 \pm 0.005$ | $0.873 \pm 0.013$ | $0.187 \pm 0.008$ | $0.841 \pm 0.008$ |
| PCGrad | $0.327 \pm 0.008$ | $0.857 \pm 0.008$ | $0.272 \pm 0.001$ | $0.828 \pm 0.003$ | $0.189 \pm 0.003$ | $0.872 \pm 0.003$ | $0.186 \pm 0.010$ | $0.830 \pm 0.006$ |
| rW | $0.329 \pm 0.008$ | $0.864 \pm 0.009$ | $0.274 \pm 0.003$ | $0.833 \pm 0.003$ | $0.186 \pm 0.003$ | $0.862 \pm 0.005$ | $0.181 \pm 0.004$ | $0.823 \pm 0.009$ |
| Linear - $\alpha = 0.25$ | $0.327 \pm 0.006$ | $0.862 \pm 0.007$ | $0.275 \pm 0.001$ | $0.835 \pm 0.005$ | $0.184 \pm 0.003$ | $0.853 \pm 0.019$ | $0.182 \pm 0.002$ | $0.825 \pm 0.007$ |
| Linear - $\alpha = 0.5$ | $0.322 \pm 0.009$ | $0.850 \pm 0.005$ | $0.272 \pm 0.001$ | $0.824 \pm 0.004$ | $0.187 \pm 0.005$ | $0.859 \pm 0.009$ | $0.184 \pm 0.006$ | $0.829 \pm 0.008$ |
| **Our Model** | **$0.340 \pm 0.005$** | **$0.913 \pm 0.019$** | **$0.291 \pm 0.003$** | **$0.905 \pm 0.004$** | **$0.200 \pm 0.007$** | **$0.916 \pm 0.008$** | **$0.197 \pm 0.003$** | **$0.892 \pm 0.007$** |

Table 10: Topic Coherence results with different pooling functions for positive set representation.

| Method | 20NG | | IMDb | | Wiki | | AGNews | |
|---|---|---|---|---|---|---|---|---|
| | $T = 50$ | $T = 200$ | $T = 50$ | $T = 200$ | $T = 50$ | $T = 200$ | $T = 50$ | $T = 200$ |
| MinPooling | $0.333 \pm 0.006$ | $0.284 \pm 0.003$ | $0.195 \pm 0.006$ | $0.191 \pm 0.010$ | $0.494 \pm 0.007$ | $0.484 \pm 0.007$ | $0.324 \pm 0.009$ | $0.211 \pm 0.001$ |
| MeanPooling | $0.335 \pm 0.003$ | $0.284 \pm 0.005$ | $0.197 \pm 0.006$ | $0.194 \pm 0.007$ | $0.492 \pm 0.009$ | $0.481 \pm 0.006$ | $0.324 \pm 0.013$ | $0.213 \pm 0.002$ |
| SumPooling | $0.336 \pm 0.006$ | $0.286 \pm 0.004$ | $0.195 \pm 0.007$ | $0.186 \pm 0.008$ | $0.493 \pm 0.009$ | $0.481 \pm 0.008$ | $0.320 \pm 0.013$ | $0.212 \pm 0.001$ |
| GAP | $0.336 \pm 0.006$ | $0.289 \pm 0.005$ | $0.196 \pm 0.006$ | $0.186 \pm 0.005$ | $0.492 \pm 0.010$ | $0.482 \pm 0.007$ | $0.318 \pm 0.009$ | $0.210 \pm 0.002$ |
| FSP | $0.336 \pm 0.004$ | $0.285 \pm 0.003$ | $0.197 \pm 0.006$ | $0.191 \pm 0.010$ | $0.494 \pm 0.012$ | $0.484 \pm 0.008$ | $0.323 \pm 0.014$ | $0.218 \pm 0.002$ |
| PSWE | $0.333 \pm 0.007$ | $0.285 \pm 0.005$ | $0.198 \pm 0.005$ | $0.187 \pm 0.006$ | $0.494 \pm 0.009$ | $0.484 \pm 0.005$ | $0.322 \pm 0.008$ | $0.211 \pm 0.002$ |
| **MaxPooling** | **$0.340 \pm 0.005$** | **$0.291 \pm 0.003$** | **$0.200 \pm 0.007$** | **$0.197 \pm 0.003$** | **$0.496 \pm 0.010$** | **$0.491 \pm 0.006$** | **$0.351 \pm 0.012$** | **$0.221 \pm 0.003$** |

The table demonstrates that our choice of MinPooling for positive set representation construction provides the most effective topic quality. We hypothesize that learning to capture similarity between strong features does not supply sufficiently hard signal for the model. As such, min-pooling will encourage topic model to focus on more difficult mutual information that can polish topic representations.

## D   ADDITIONAL EVALUATION OF TOPIC QUALITY

We investigate a larger scope for our proposed setwise contrastive topic model, where we train prior baselines and our model on Yahoo Answer (Zhang et al., 2015) dataset, which consists of about $1.4$ million training and testing documents. The illustrated results in Table 11 demonstrate that setwise contrastive topic model does polish the produced topics better than other topic models, proving both of its effectiveness and efficacy.

## E   EXAMPLES OF DATA AUGMENTATION METHODS

In this section, we llustrate the examples generated by our augmentation, TFIDF-based, and embedding-based in Table 12, 13, and 14, respectively. For embedding-based augmentation, we use Word2Vec (Mikolov et al., 2013) as the pretrained word embedding embedding.

## F   STATISTICAL SIGNIFICANCE TESTS TOWARDS TOPIC QUALITY

In this appendix, to more rigorously assess the quality of our generated topics compared with baseline models, we conduct a statistical t-test for performance comparison in Section 5. In particular, we select the second-best method in all Tables, *i.e.* NTM+CL, Linear-$\alpha = 0.5$, NTM+CL, NTM+CL, GAP, and GAP in Table 4, 5, 2, 3, 6, and 10, respectively. We provide our obtained $p$-values in Table 15, 16, 17, 18, 19, and 20. The results show that our improvement is statistically significant with all $p$-value $< 0.05$.

Table 11: Topic Coherence and Topic Diversity results on Yahoo Answers dataset.

| Method | Yahoo Answers | | | |
|---|---|---|---|---|
| | $T = 50$ | | $T = 200$ | |
| | NPMI | TD | NPMI | TD |
| NTM | $0.290\pm0.003$ | $0.897\pm0.012$ | $0.161\pm0.001$ | $0.718\pm0.004$ |
| ETM | $0.207\pm0.001$ | $0.808\pm0.017$ | $0.158\pm0.002$ | $0.709\pm0.007$ |
| DVAE | $0.319\pm0.001$ | $0.932\pm0.014$ | $0.163\pm0.003$ | $0.721\pm0.004$ |
| BATM | $0.247\pm0.004$ | $0.883\pm0.019$ | $0.177\pm0.001$ | $0.767\pm0.003$ |
| W-LDA | $0.222\pm0.004$ | $0.821\pm0.008$ | $0.173\pm0.005$ | $0.730\pm0.001$ |
| SCHOLAr | $0.327\pm0.005$ | $0.936\pm0.019$ | $0.191\pm0.003$ | $0.784\pm0.004$ |
| SCHOLAr+BAT | $0.328\pm0.005$ | $0.940\pm0.020$ | $0.196\pm0.001$ | $0.793\pm0.005$ |
| NTM+CL | $0.334\pm0.002$ | $0.954\pm0.004$ | $0.198\pm0.001$ | $0.794\pm0.003$ |
| **Our Model** | $\mathbf{0.348}\pm\mathbf{0.004}$ | $\mathbf{0.988}\pm\mathbf{0.003}$ | $\mathbf{0.208}\pm\mathbf{0.002}$ | $\mathbf{0.882}\pm\mathbf{0.001}$ |

Table 12: Examples generated by our augmentation.

| Input document x | Our method | |
|---|---|---|
| | $\mathbf{x}^+$ | $\mathbf{x}^-$ |
| ronaldo poaches the points as owen tastes first real. | ronaldo's scoring prowess was on full display as he secured the victory for real, while owen experienced his first taste of success with the team. | the serene sunset over the ocean painted the sky with hues of orange and pink, casting a peaceful aura over the coastal town. |
| i quite often find that when a film doesn't evolve around a famous actor or actress but rather a story or style. | focusing on a compelling story or unique cinematic style can often lead to a more engaging movie experience, regardless of the star power of the actors involved. | the cat jumped over the fence and landed gracefully in the neighbor's yard. |
| finally, a way to Google your hard drive You can usually find information more easily on the vast internet. | with this new tool, you'll be able to search your hard drive just as efficiently as you browse the vast expanse of the internet. | the sound of laughter echoed through the empty park. |
| bioshield effort is inadequate, a study says experts say the bioshield program will not be adequate to create new vaccines and drugs to protect the nation against either a biological attack or natural epidemic. | the study's findings cast doubt on the effectiveness of the bioshield program in developing the necessary vaccines and drugs to safeguard the nation from biological threats, according to experts. | the chef carefully placed the garnish on the plate, completing the presentation of the dish. |

## G  DYNAMICS OF $\alpha$ IN MULTI-OBJECTIVE OPTIMIZATION

We inspect the change of alpha during training to better understand the role of our setwise contrastive learning for neural topic model. Figure 3 demonstrates that in the beginning, alpha is fairly small, which indicates that initially focusing on the ELBO objective of neural topic model is more effective. Subsequently, the value of alpha increases, denoting that contrastive learning gradually affects the training. This can be considered as an evidence that validates the effect of our setwise contrastive learning on neural topic model.

## H  HUMAN EVALUATION: HUMAN RATINGS AND WORD INTRUSION TASKS

Since human judgement towards the generated topics might deviate from the results of automatic metrics (Hoyle et al., 2021), we decide to conduct human evaluation on the 20NG and IMDb datasets through utilizing two tasks, *i.e.* human ratings and word intrusion task. Our evaluation is conducted upon Amazon Mechanical Turk (AMT). In particular, for human ratings, we ask annotators to rate the generated topics on the scale from 1 to 3. For word intrusion task, in each topic, we insert an outsider word and ask human annotators to select which word is the outsider word. Our human evaluation is performed on our model and the NTM+CL model of (Nguyen & Luu, 2021). We record the human evaluation results and plot them in Figure 4. As can be observed, human judgements prefer our

Table 13: Examples generated by TFIDF-based augmentation (Nguyen & Luu, 2021).

| Input document x | TFIDF-based | |
| | $x^+$ | $x^-$ |
| --- | --- | --- |
| ronaldo poaches the points as owen tastes first real. | cristiano cambone this rebounds one bellamy distinctly fourth gives. | maradona lem following 18 own ritchie mixes 1998 rather. |
| i quite often find that when a film doesn't evolve around a famous actor or actress but rather a story or style. | never seemed ones way has he with screenplay reason we alter apart with notable film rather girlfriend only meant with explains rather simple. | get rarely perhaps still actually turn put episode come nothing conversely instead put besides scriptwriter there ms. while instance put inspired there quirky. |
| finally, a way to Google your hard drive You can usually find information more easily on the vast internet. | once both with it move online get turning stopping maybe if instead way account still quickly which this wealth websites. | turned instead put once rest lets want lot hard lot these less still link longer none last following boasts msn. |
| bioshield effort is inadequate, a study says experts say the bioshield program will not be adequate to create new vaccines and drugs to protect the nation against either a biological attack or natural epidemic. | nobelity helping part sufficiently both with suggests explained researchers know this nobelity planning ready any even providing move rather its doses made treatment move threatened this entire losing however with possessed fire rather unique malaria. | circumvents promises once poorly instead put applied nothing impact n't following circumvents expanded step thought cannot suitable rest core introduced medication few antiviral rest need following strongest twice as put analyzing enemy there hence spreads. |

Table 14: Examples generated by embedding-based augmentation (Yin & Schütze, 2016).

| Input document x | Embedding-based | |
| | $x^+$ | $x^-$ |
| --- | --- | --- |
| ronaldo poaches the points as owen tastes first real. | ronaldinho strafaci which point also keane taste second kind. | artd newsfeatures yahd nzse40 404-526-5456 3,399 leel arabised highjump. |
| i quite often find that when a film doesn't evolve around a famous actor or actress but rather a story or style. | me very sometimes how but then an movie if know evolving where an famed starred instead actor though merely an book instead styles. | statistique kunzig -96 deylam suhn micale lacques nzse40 xinglong myoo fdch lacques -5000 basse-normandie zahr debehogne cetacea lacques arabised zahr tih. |
| finally, a way to google your hard drive you can usually find information more easily on the vast internet. | again . an so take yahoo my turn drives 'll able typically how source than either . which surrounding web. | sahk lacques -96 yahd raht 1-732-390-4480 milanka musicnotes.com 1-732-390-4480 photoexpress suhn deylam tahn 100.00 zhahn romanosptimes.com newsfeatures puh condita. |
| bioshield effort is inadequate, a study says experts say the bioshield program will not be adequate to create new vaccines and drugs to protect the nation against either a biological attack or natural epidemic. | toshka efforts this insufficient . an studies thinks specialists believe which toshka programs would be not sufficient take creating for vaccine well drug take protecting which country opponents instead an chemical attacks instead abundant outbreak. | chivenor wmorrisglobe.com nehn europe/africa sahk lacques ortsgemeinden telegram.com gahl e-bangla newsfeatures chivenor sholdersptimes.com ortsgemeinden milanka 47-42-80-44 sylviidae yahd troo yanow chron.com bizmags tilove yahd telegram.com newsfeatures juh speech/language tvcolumn lacques fassihi 2378 zahr berris bse-100. |

generated topics to the topics of NTM+CL (Nguyen & Luu, 2021). The inter-annotator agreement is 0.72, which indicates high agreement among human annotators.

Table 15: Statistical test for document classification results using topic representations of Our model and NTM+CL.

| Metric | 20NG | IMDb | AGNews |
|---|---|---|---|
| $p$-value | 0.048 | 0.023 | 0.019 |

Table 16: Statistical test for the ablation study on objective control strategy of Our model and Linear-$\alpha = 0.5$.

| Metric | Wiki | |
|---|---|---|
| | $T = 50$ | $T = 200$ |
| $p$-value | 0.010 | 0.039 |

Table 17: Statistical test for the comparison of the topic coherence and topic diversity results between Our model and NTM+CL on 20NG and IMDb datasets.

| Metric | 20NG | | | | IMDb | | | |
|---|---|---|---|---|---|---|---|---|
| | $T = 50$ | | $T = 200$ | | $T = 50$ | | $T = 200$ | |
| | NPMI | TD | NPMI | TD | NPMI | TD | NPMI | TD |
| $p$-value | 0.034 | 0.032 | 0.047 | 0.047 | 0.024 | 0.029 | 0.015 | 0.024 |

Table 18: Statistical test for the comparison of the topic coherence and topic diversity results between Our model and NTM+CL on Wiki and AG News datasets.

| Metric | Wiki | | | | AG News | | | |
|---|---|---|---|---|---|---|---|---|
| | $T = 50$ | | $T = 200$ | | $T = 50$ | | $T = 200$ | |
| | NPMI | TD | NPMI | TD | NPMI | TD | NPMI | TD |
| $p$-value | 0.037 | 0.044 | 0.005 | 0.018 | 0.033 | 0.047 | 0.029 | 0.026 |

Table 19: Statistical test for the comparison of pooling functions for negative samples between our MaxPooling and the GAP method.

| Metric | 20NG | | IMDb | | Wiki | | AGNews | |
|---|---|---|---|---|---|---|---|---|
| | $T = 50$ | $T = 200$ | $T = 50$ | $T = 200$ | $T = 50$ | $T = 200$ | $T = 50$ | $T = 200$ |
| $p$-value | 0.044 | 0.013 | 0.013 | 0.017 | 0.001 | 0.005 | 0.008 | 0.019 |

Table 20: Statistical test for the comparison of pooling functions for positive samples between our MinPooling and the GAP method.

| Metric | 20NG | | IMDb | | Wiki | | AGNews | |
|---|---|---|---|---|---|---|---|---|
| | $T = 50$ | $T = 200$ | $T = 50$ | $T = 200$ | $T = 50$ | $T = 200$ | $T = 50$ | $T = 200$ |
| $p$-value | 0.017 | 0.033 | 0.004 | 0.047 | 0.030 | 0.036 | 0.011 | 0.003 |

Figure 3: Dynamics of $\alpha$ during training on four datasets 20NG, IMDb, Wiki, and AGNews with $T = 50$.

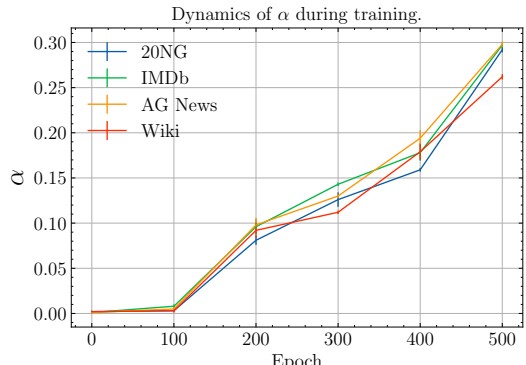

Figure 4: Human evaluation on 20NG and IMDb datasets with two tasks: human ratings and word intrusion.

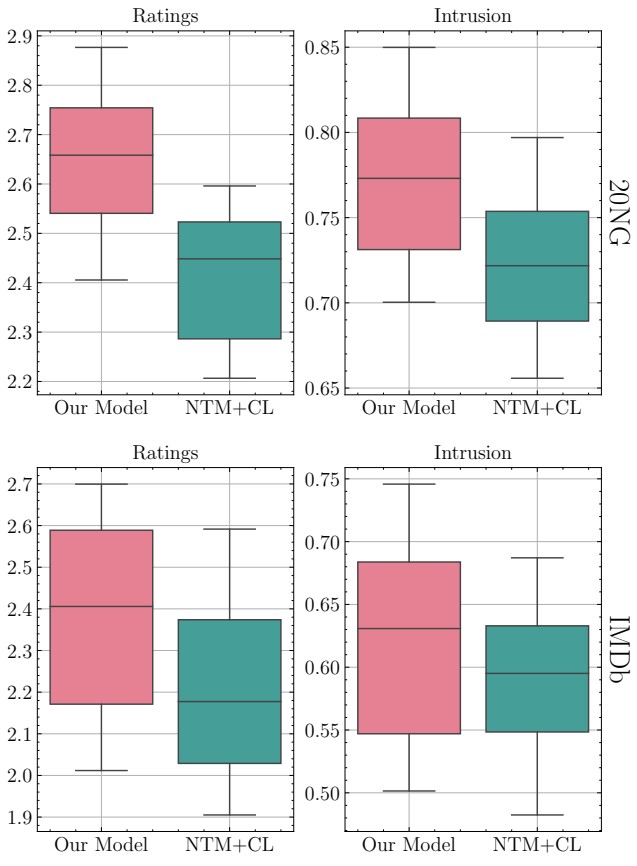

