# OpenReview forum: "Topic Modeling as Multi-Objective Contrastive Optimization"
_ICLR.cc/2024/Conference — ICLR 2024 poster_

### Official Review · Reviewer_QHKf · 2023-10-30

**Soundness:** 3 good
**Presentation:** 2 fair
**Contribution:** 3 good
**Rating:** 6
**Confidence:** 3

**Summary:**

This paper proposes a set-wise contrastive learning algorithm for neural topic models, where the input document batch is divided into sets and positives and negatives are constructed by augmenting and pooling instances in each set. It casts the proposed method as a multi-objective optimization problem to balance the trade-off between the ELBO and the contrastive objectives. The ChaptGPT API is used for data augmentation to generate negative samples. Experiments demonstrate the effectiveness of the proposed method.

**Strengths:**

- The set-wise contrastive learning is new and effective for resolving low-level mutual information of neural topic models.
- Formulating the contrastive learning as a multi-task learning problem and solving it by a multi-objective optimization algorithm is an interesting idea.
- Experimental comparisons with state-of-the-art neural topic models that include recent ones such as WeTe demonstrate the effectiveness of the proposed moethod.

**Weaknesses:**

- The effectiveness of the ChatGPT-based data augmentation is unclear.
- The formal definition of the contrastive loss is  missing in the main text, while incomplete definition can be found in Algorithm 1.
- The justification of the use of MaxPooling is unclear.
- The authors seem to assume as if it is possible to find "the optimal" Pareto solution (a Pareto optimal solution with optimal balance), while there is no superiority or inferiority between Pareto optimal solutions.

**Questions:**

In Section 4.1, the authors adopt ChatGPT API to perform the augmentation with the specific prompt.
However, at the end of Section 5.1, they use Word2Vec as pretrained embedding for their embedding-based augmentation.
This is confusing. Do they perform both ChatGPT-based and embedding-based augmentations for comparisons?
Anyway, they should show the effectiveness of the ChatGPT-based augmentation.

In Section 5.5 on page 9, they justify the use of MaxPooling, for extracting set representation,  as MaxPooling directly retrieving strong features. In Table 6, they compare different pooling functions including max and min poolings.
However, in Algorithm 1, both MaxPool and MinPool are used. Here, MinPool is used for positive pair.
They should clarify the reason why.

The purpose of multi-objective optimization is to find a  (possibly diversified) set of Pareto optimal solutions so that it represents the Pareto frontier. Therefore, I would like to know the average and standard deviation of the $\alpha$  obtained at the end of each run to see how it is diversified. If it is not diversified, then I would like to know the reason.

Judging from the definition of $\mathcal{L}_{set}$ in Algorithm 1, both the positive and the negative pairs are generated within the same set and the loss is then accumulated for all the sets. If this is the case, then the first sentence of the Hard Negative Sampling paragraph sounds confusing. Differenciating among sets may be more difficult, but the contrastive pair for differenciating is generated within a set not among sets.  In Algorithm 1,  $\exp(s_i^{min}, s_i^{+})$ should be $\exp(f(s_i^{min}, s_i^{+}))$.

How and what value for the contrastive weight $\beta$ is determined?

**Details Of Ethics Concerns:**

I have no concerns.

---

> ### Author Response · Authors · 2023-11-15
> **Thank you for your helpful reviews**
>
> Thank you for your helpful reviews! We are happy that you appreciate our novel, effective, and interesting method. We sincerely hope that our response and the rebuttal version of our paper can address your concerns and improve your ratings.
>
> **Q1: Do you perform both ChatGPT-based and embedding-based augmentations for comparisons?**
>
> We use Word2Vec to generate augmentation examples in Appendix H. In our main experiments in section 5.2 - 5.3, we only use ChatGPT generated examples. **We have moved the Word2vec detail to Appendix H to make our work clearer.**
>
> **Q2: Reason why both MinPool and MaxPool are used.**
>
> In Algorithm 1, we use max pooling for negative samples because we want to focus on pushing away strongest features of negative samples and suppress their noise. We use min pooling for positive samples because we want to attract even the weakest features of positive samples.
>
> In section 5.5, we experiment with the pooling function for negative samples. We have compared pooling function choices for positive samples:
>
> |             |    20NG   |           |    IMDb   |           |    Wiki   |           |   AGNews  |           |
> |-------------|:---------:|:---------:|:---------:|:---------:|:---------:|:---------:|:---------:|:---------:|
> | **Method**  |  $T = 50$ | $T = 200$ |  $T = 50$ | $T = 200$ |  $T = 50$ | $T = 200$ |  $T = 50$ | $T = 200$ |
> | MaxPooling  |   0.333   |   0.284   |   0.195   |   0.191   |   0.494   |   0.484   |   0.324   |   0.211   |
> | GAP         |   0.336   |   0.289   |   0.196   |   0.186   |   0.492   |   0.482   |   0.318   |   0.210   |
> | FSP         |   0.336   |   0.285   |   0.197   |   0.191   |   0.494   |   0.484   |   0.323   |   0.218   |
> | MinPooling  | **0.340** | **0.291** | **0.200** | **0.197** | **0.496** | **0.491** | **0.351** | **0.221** |
>
> We can see that our min pooling provides better aggregated mutual information for positive samples in neural topic modeling. **We have added these details in Section 5.5 and Table 10 in Appendix F of the rebuttal revision.**
>
> **Q3: The average and standard deviation of $\alpha$.**
>
> We observe that the values of $\alpha$ vary from the starting epoch to the final epoch. In particular, in the beginning, the mean and standard deviation of alpha are about 0.001. This means that in the beginning, it is better to focus on optimizing the ELBO objective of the neural topic model.
>
> Gradually the value of $\alpha$ increases and converges to approximately 0.3 in the last epoch. This validates the usefulness of our setwise contrastive learning that gives beneficial mutual information signal to topic modeling.
>
> We specify the values of $\alpha$ after epochs for 20NG and IMDb datasets in the following table. **We have added $\alpha$ values for all other datasets to Appendix J in our rebuttal revision.**
>
> | |              |         |            |          |          |          |
> |---------------|:-------------:|:-------------:|:-------------:|:-------------:|:-------------:|:-------------:|
> | **Dataset/Epoch** |       **0**      |      **100**      |      **200**      |      **300**      |      **400**     |      **500**      |
> | 20NG          | 0.002 ± 0.001 | 0.003 ± 0.001 | 0.081 ± 0.005 | 0.126 ± 0.008 | 0.159 ± 0.002 | 0.292 ± 0.003 |
> | IMDb          | 0.001 ± 0.001 | 0.008 ± 0.002 | 0.096 ± 0.008 | 0.143 ± 0.002 | 0.178 ± 0.009 | 0.297 ± 0.003 |
>
> **Q4: The first sentence of the Hard Negative Sampling paragraph**
>
> Sincerely thank you for your note. We have provided two modifications for Algorithm 1 in our rebuttal revision to more accurately describe our method:
>
> - Include the iteration $p$ that loops over the permutation matrix: We diversify our document sets by permuting the document index within a batch. **We have added the index $p$ to the formula using MaxPool and MinPool operations to reflect this (line 10-13 in Algorithm 1)**.
>
> - Modify the contrastive loss: In our implementation, we consider a negative pair as a document set $i$ with all other document sets $j$ which are generated based on the permutation matrix $M$ (except for the set which includes corresponding augmented documents of set $i$). **We have corrected the denominator of our contrastive loss to reflect this (line 16 in Algorithm 1).**
>
> Therefore, as in the first sentences of our hard negative sampling paragraph, we put a document into multiple sets and compare among them. This would create hard negative samples and benefit contrastive learning.
>
> **Q5: In Algorithm 1, $\text{exp}(s_{i}^{min}, s_{i}^{+})$ should be $\text{exp}(f(s_{i}^{min}, s_{i}^{+}))$**
>
> Thank you so much for your note. **We have edited this detail in Algorithm 1 (line 16) of our rebuttal revision.**
>
> **Q6: The value of contrastive weight $\beta$.**
>
> We compare between the adaptive scheduling in NTM+CL (Nguyen et al., 2021) and fixing it to 1. However, since the difference is not significant, for simplicity we directly set $\beta$ to 1 (line 16 in Algorithm 1).

---

> > ### Author Response · Authors · 2023-11-20
> > **Looking forward to your insightful feedback**
> >
> > Thank you sincerely for your helpful reviews!
> >
> > We mention that we have submitted responses including **clarifications for embedding-based augmentation, pooling functions, $\alpha$ values, and Algorithm 1.**
> >
> > We hope our responses can address your concerns and improve your rating.
> >
> > We are looking forward to your insightful feedback!
> >
> > Thank you for your help!

---

> > ### Comment · Reviewer_QHKf · 2023-11-20
> >
> > Thank you for the clarifications.
> >
> > Concerning Q4 and Q5, I do hope you improve the description of Algorithm 1 for a better presentation.
> > Besides, the last line should be $\phi = \phi - \dots$.

---

> > > ### Author Response · Authors · 2023-11-20
> > > **Thank you for your quick and insightful response**
> > >
> > > Thank you for your quick and insightful response! We have added the comments and rewritten the formula (lines 9-14) in Algorithm 1 to make our algorithm clearer and more coherent with our description (Section 4.1, Section 5.5, and Appendix F). We look forward to hearing your feedback soon.

---

> > > ### Author Response · Authors · 2023-11-23
> > >
> > > Dear Reviewer QHKf,
> > >
> > > We wish to express our sincere gratitude once again to you for the valuable contributions and considerate feedback. We would like to gently bring to your attention that the discussion phase between authors and reviewers is nearing completion (within 6 hours).
> > >
> > > We have added description to Algorithm 1 to clarify the setwise contrastive learning method and make our contribution clearer. We kindly hope our responses can address your concerns and improve your rating. Should you have any further insights to share, we are more than willing to sustain our discussion until the deadline.

---

> > > > ### Comment · Reviewer_QHKf · 2023-11-23
> > > >
> > > > I appreciate the authors' efforts to correctly rewrite the manuscript. Assuming everything is now consistent in the revised manuscript, I would like to keep my score as it is in a good sense.

---

### Official Review · Reviewer_3szg · 2023-10-31

**Soundness:** 3 good
**Presentation:** 3 good
**Contribution:** 3 good
**Rating:** 6
**Confidence:** 3

**Summary:**

This paper proposes a setwise contrastive learning algorithm for neural topic model to address the problem of learning low-level mutual information of neural topic models. This work explicitly casts contrastive topic modeling as a gradient-based multi-objective optimization problem. Extensive experiments are performed to demonstrate the effectiveness of this method.

**Strengths:**

1. This paper is well-organized and equations are clearly written.
2. Extensive experimental are performed and results show this method consistently presents high performance.
3. Codes are provided in supplementary materials to ensure reproducibility.

**Weaknesses:**

1. Since [1] also uses contrastive learning to capture useful semantics of topic vectors which is similar to the proposed method, this paper does not clearly compare with [1] and explain its novelty.
2. This paper omits important baselines. For example, [1] also presents great performance in this task but this paper does not compare with it in the experiments. Which contrastive learning method performs better?

[1] Xiaobao Wu, Anh Tuan Luu, and Xinshuai Dong. Mitigating data sparsity for short text topic modeling by topic-semantic contrastive learning. arXiv preprint arXiv:2211.12878, 2022.

**Questions:**

1. I have a question about the baselines. Table 5 shows topic diversity (TD) score of WeTe is 0.878±0.012 when T = 50 in AG News dataset. However, in [1], this score is 0.966 without data augmentation and 0.991 with data augmentation. There is a large gap between these scores. I wonder how authors implement these baselines. Could you please provide more details or any codes?

[1] Xiaobao Wu, Anh Tuan Luu, and Xinshuai Dong. Mitigating data sparsity for short text topic modeling by topic-semantic contrastive learning. arXiv preprint arXiv:2211.12878, 2022.

---

> ### Author Response · Authors · 2023-11-15
> **Thank you for your insightful reviews**
>
> Thank you for your insightful reviews! We are delighted that you appreciate our clear, reasonable method, and our rigorous experiments. We sincerely hope our response and the rebuttal version of our paper can address your questions and improve your ratings.
>
> **Q1: Details about topic diversity**
>
> We clarify that to measure the topic diversity, TSCTM [1] and WeTe [2] extract 25 top words in each topic, while we only use 10 top words, which would make the evaluation harder. If we follow strictly follow their protocol, most of the topic diversity values will be ~0.999, which cannot discriminatively evaluate the topic models.
>
> **Q2: Comparison with (Wu et al., 2022 [1]) work**
>
> We clarify that the main difference between our work and TSCTM (Wu et al., 2022 [1]) lies in that their work is restricted to short texts, *e.g.* TagMyNews dataset (mean length: 8 words, max length: 17 words), and GoogleNews (mean length: 5.38 words, max length: 12 words). In contrast, we work with texts of multiple lengths, *e.g.* AGNews (mean length: 20.19 words, max length: 76 words), 20NG (mean length: 152.21 words, max length: 12419 words), and Wiki (mean length: 130.14 words, max length: 640 words).
>
> We also compare our model with TSCTM in the settings of $T = 50$ topics:
>
> |            |   20NG   |        |   IMDb   |        |   Wiki   |        |  AGNews  |        |
> |------------|:--------:|:------:|:--------:|:------:|:--------:|:------:|:--------:|:------:|
> | **Method** | **NPMI** | **TD** | **NPMI** | **TD** | **NPMI** | **TD** | **NPMI** | **TD** |
> | TSCTM      |   0.271  |  0.668 |   0.149  |  0.741 |   0.327  |  0.753 |   0.252  |  0.783 |
> | Our Model  |   0.340  |  0.913 |   0.291  |  0.916 |   0.496  |  0.959 |   0.351  |  0.934 |
>
> We see that our model outperforms TSCTM on both topic diversity and topic coherence. We believe the reason is that our setwise contrastive learning captures more mutual information than TSCTM in documents of various lengths. Thank you for your comment. **We’ve added these to Table 4 and Table 5 of our rebuttal revision.**
>
> **References:**
>
> [1] Wu et al., Mitigating data sparsity for short text topic modeling by topic-semantic contrastive learning, arXiv 2022.
>
> [2] Wang et al., Representing mixtures of word embeddings with mixtures of topic embeddings, arXiv 2022.

---

> ### Author Response · Authors · 2023-11-23
>
> Dear Reviewer 3szg,
>
> We wish to express our sincere gratitude once again to you for the valuable contributions and considerate feedback. We would like to gently bring to your attention that the discussion phase between authors and reviewers is nearing completion (within 6 hours).
>
> Given the inclusion of **the comparison results with related works and our clarification towards topic diversity**, we kindly hope our responses can address your concerns and improve your rating. Should you have any further insights to share, we are more than willing to sustain our discussion until the deadline.

---

> > ### Comment · Reviewer_3szg · 2023-11-23
> >
> > Thanks for the response of the authors. I appreciate the efforts the authors made to address my concern on the experiments. This part looks better now and I would like to raise my rating from 5 to 6. For the novelty and motivation explanation, I would like to leave it to the AC to decide whether to accept.

---

> > > ### Author Response · Authors · 2023-11-23
> > >
> > > Thank you for your response! We are happy that you appreciate our effort to address your concerns.

---

### Official Review · Reviewer_ioHf · 2023-11-03

**Soundness:** 2 fair
**Presentation:** 4 excellent
**Contribution:** 3 good
**Rating:** 6
**Confidence:** 3

**Summary:**

This paper aims to address challenges in neural topic models (NTMs) caused by recent approaches that optimize the combination of ELBO and contrastive learning. These challenges include capturing low-level mutual information and a conflict between ELBO's focus on input details for reconstruction quality and contrastive learning's goal of generalizing topic representations. To overcome these issues, the authors propose a novel setwise contrastive learning method for sets of documents, aiming to capture shared semantics among documents. Additionally, they formulate contrastive topic modeling as a multi-objective optimization problem to achieve a balanced solution. Experimental results on 4 benchmark datasets demonstrate that their approach consistently produces higher-performing NTMs in terms of topic coherence, diversity, and document classification performance compared to existing methods.

**Strengths:**

+ The key idea of this paper (i.e., learning low-level mutual information of neural topic models that optimize ELBO and contrastive learning together) is very well-motivated.

+ The usage of setwise contrastive topic modeling is reasonable. Casting it as a multi-task learning problem and adopting multi-objective optimization to find a Pareto solution are technically novel.

+ A comprehensive set of benchmark datasets and baselines are considered. The authors also perform detailed ablation studies, hyperparameter analyses, and case studies.

**Weaknesses:**

- Statistical significance tests are missing. It is unclear whether the gaps between the proposed model and baselines/ablation versions are statistically significant or not. In particular, some gaps in Tables 3 and 6 are quite subtle, and the variances of classification scores in Table 2 are unknown, therefore p-values should be reported.

- Only automatic metrics (e.g., NPMI and TD) are used to evaluate topic quality. Although the authors also examine document classification as a downstream task, the classification performance is just an indirect measurement of topic quality. Recent work [1] has shown that automatic metrics may deviate from humans' judgment. Therefore, I feel human evaluation (e.g., the intrusion test [2]) is still needed.

[1] Is Automated Topic Model Evaluation Broken?: The Incoherence of Coherence. NeurIPS'21.

[2] Topic Intrusion for Automatic Topic Model Evaluation. EMNLP'18.

**Questions:**

- Could you conduct statistical significance tests to compare your method with the baselines in the experiment tables?

- Could you perform human evaluation (e.g., the intrusion test) to directly examine the quality of extracted topics?

---

> ### Author Response · Authors · 2023-11-15
> **Thank you for your helpful reviews**
>
> Thank you for your helpful reviews! We are happy that you appreciate our well-motivated and novel method. We sincerely hope our response and the rebuttal version of our paper can address your concerns and improve your ratings.
>
> **Q1: p-values should be reported.**
>
> Thank you for your suggestion. We have conducted a t-test between our method and the second-best method in Table 2-6, i.e. NTM+CL, Linear-$\alpha=0.5$, NTM+CL, NTM+CL, and GAP, respectively. We obtain $p$-values as follows:
>
> - **t-test for Table 2:**
>
> | Metric  | 20NG  | IMDb  | AGNews |
> |---------|-------|-------|--------|
> | $p$-value | 0.048 | 0.023 | 0.019  |
>
> - **t-test for Table 3:**
>
> | |  Wiki  | |
> |---------|:------:|:-------:|
> |    **Metric**       | $T = 50$ | $T = 200$ |
> | $p$-value |  0.010 |  0.039  |
>
> - **t-test for Table 4:**
>
> |         |  20NG  |       |         |       |  IMDb  |       |         |       |
> |---------|:------:|:-----:|:-------:|:-----:|:------:|:-----:|:-------:|:-----:|
> |         | $T = 50$ |       | $T = 200$ |       | $T = 50$ |       | $T = 200$ |       |
> | **Metric**  |  **NPMI**  |   **TD**  |   **NPMI**  |   **TD**  |  **NPMI**  |   **TD**  |   **NPMI**  |   **TD** |
> | $p$-value |  0.034 | 0.032 |  0.047  | 0.047 |  0.024 | 0.029 |  0.015  | 0.024 |
>
> - **t-test for Table 5:**
>
> |         |  Wiki  |       |         |       | AGNews |       |         |       |
> |---------|:------:|:-----:|:-------:|:-----:|:------:|:-----:|:-------:|:-----:|
> |         | $T = 50$ |       | $T = 200$ |       | $T = 50$ |       | $T = 200$ |       |
> | **Metric**  |  **NPMI**  |   **TD**  |   **NPMI**  |   **TD**  |  **NPMI**  |   **TD**  |   **NPMI**  |   **TD** |
> | $p$-value |  0.037 | 0.044 |  0.005  | 0.018 |  0.033 | 0.047 |  0.029  | 0.026 |
>
> - **t-test for Table 6:**
>
> |            |   20NG   |           |   IMDb   |           |   Wiki   |           |  AGNews  |           |
> |:----------:|:--------:|:---------:|:--------:|:---------:|:--------:|:---------:|:--------:|:---------:|
> | **Metric** | $T = 50$ | $T = 200$ | $T = 50$ | $T = 200$ | $T = 50$ | $T = 200$ | $T = 50$ | $T = 200$ |
> | $p$-value  |   0.044  |   0.013   |   0.013  |   0.017   |   0.001  |   0.005   |   0.008  |   0.019   |
>
> We see that all of the results are statistically significant ($p$-value < 0.05). We believe the reason is that our method better refines the topic representation through setwise contrastive signal and our effective Pareto solution for the NTM. Thank you again for your advice. **We have added these to Appendix I in our rebuttal revision.**
>
> **Q2: Human evaluation is needed.**
>
> Thank you for your comment. We have followed previous works [1, 2, 3] to utilize two automatic metrics, *i.e.* topic diversity and NPMI, to measure topic quality. However, we acknowledge that human evaluation will provide a more rigorous evaluation. We have conducted human rating and word intrusion tasks for 20NG and IMDb on Amazon Mechanical Turk (AMT) between our topics and those of NTM+CL, and **obtain the result in Figure 5 in Appendix K of our rebuttal revision.**
>
> We see that human judgement also prefers our generated topics than the baseline ones. Our inter-annotator agreement is 0.72, which indicates high agreement. Thank you for your interesting suggestion, **we have added these to Appendix K in our rebuttal revision.**
>
> **References:**
>
> [1] Thong Nguyen and Anh Tuan Luu, Contrastive learning for neural topic model, NeurIPS 2021.
>
> [2] Xu et al., Hyperminer: Topic taxonomy mining with hyperbolic embedding, NeurIPS 2022.
>
> [3] Wang et al., Representing mixtures of word embeddings with mixtures of topic embeddings, arXiv 2022.

---

> ### Author Response · Authors · 2023-11-23
>
> Dear Reviewer ioHf,
>
> We wish to express our sincere gratitude once again to you for the valuable contributions and considerate feedback. We would like to gently bring to your attention that the discussion phase between authors and reviewers is nearing completion, **which is within 1 hour.**
>
> Given the inclusion of **the results for statistical significance tests and human evaluation**, we kindly hope our responses can address your concerns and improve your rating. Should you have any further insights to share, we are more than happy to sustain our discussion until the deadline.

---

### Author Response · Authors · 2023-11-22

Thank you for reading our response!

We thank the AC and all reviewers for their great efforts to review our paper. We are happy that reviewer ioHf finds our method novel and reasonable, reviewer QHKf finds our work interesting and well-motivated, and reviewer 3szg believes extensive experiments are performed and the equations are clearly written.

We’d like to emphasize that we have submitted our responses providing **results for significance tests, human evaluation, and clarifying description for our Algorithm 1 to conduct setwise contrastive learning for neural topic model.** We also note that we’ve uploaded our code, and we are ready to release it for reproducibility.

Since the discussion period is closing soon, we look forward to receiving feedback from reviewers. Also, we hope ACs could engage reviewers to join the rebuttal so that we can continue discussion for better clarification of our work.

---

### Meta-Review · Area_Chair_WRit · 2023-12-12

**Metareview:**

This paper proposes a setwise contrastive learning algorithm for neural topic models (NTMs) that addresses mutual information capture and balances the trade-off between evidence lower bound (ELBO) optimization and contrastive learning through a multi-objective optimization approach. The paper is well-received by all reviewers, with commendations for the novelty and technical soundness of the proposed method, as well as the rigorous experimental evaluation and reproducibility afforded by the provided codes.

The primary concerns raised by the reviewers revolved around the statistical significance of results, the lack of human evaluation for topic quality, clear differentiation from related work, and technical details such as the use of various pooling strategies and data augmentation methods. The authors’ rebuttal satisfactorily addresses these concerns by providing additional statistics, such as p-values, and including human evaluation results while clarifying the differences from prior work and giving more details on their methodology.

The reviewers note improvements in the paper post-rebuttal, especially with new comparisons to related work and human evaluation. Additionally, they acknowledge the authors' efforts to address all questions, although some reviewers suggest further improvements in the manuscript for clarity. In light of the detailed responses and updates made by the authors, I believe the paper is marginally above the acceptance bar. The authors have displayed a commendable effort towards incorporating feedback, showing improvements in statistical testing, human evaluation, and a more thorough comparison with related work, which has, in turn, positively influenced the ratings from the reviewers.

**Justification For Why Not Higher Score:**

The primary concerns raised by the reviewers revolved around the statistical significance of results, the lack of human evaluation for topic quality, clear differentiation from related work, and technical details such as the use of various pooling strategies and data augmentation methods. The authors’ rebuttal satisfactorily addresses these concerns by providing additional statistics, such as p-values, and including human evaluation results while clarifying the differences from prior work and giving more details on their methodology.

**Justification For Why Not Lower Score:**

The reviewers note improvements in the paper post-rebuttal, especially with new comparisons to related work and human evaluation. Additionally, they acknowledge the authors' efforts to address all questions, although some reviewers suggest further improvements in the manuscript for clarity.

---

### Decision · Program_Chairs · 2024-01-16

Accept (poster)